# Food Applications and Potential Health Benefits of Hawthorn

**DOI:** 10.3390/foods11182861

**Published:** 2022-09-15

**Authors:** Juan Zhang, Xiaoyun Chai, Fenglan Zhao, Guige Hou, Qingguo Meng

**Affiliations:** 1Department of Key Laboratory of Molecular Pharmacology and Drug Evaluation, School of Pharmacy, Yantai University, Yantai 264005, China; 2Department of Organic Chemistry, School of Pharmacy, Naval Medical University, Shanghai 200433, China; 3School of Pharmacy, Binzhou Medical University, Yantai 264003, China

**Keywords:** nutrition, flavonoids, phenolic acid, antioxidant, anti-cardiovascular disease, bread, beverage, meat

## Abstract

Hawthorn (*Crataegus*) is a plant of the Rosaceae family and is widely grown throughout the world as one of the medicinal and edible plants, known as the “nutritious fruit” due to its richness in bioactive substances. Preparations derived from it are used in the formulation of dietary supplements, functional foods, and pharmaceutical products. Rich in amino acids, minerals, pectin, vitamin C, chlorogenic acid, epicatechol, and choline, hawthorn has a high therapeutic and health value. Many studies have shown that hawthorn has antioxidant, anti-inflammatory, anticancer, anti-cardiovascular disease, and digestive enhancing properties. This is related to its bioactive components such as polyphenols (chlorogenic acid, proanthocyanidin B2, epicatechin), flavonoids (proanthocyanidins, mucoxanthin, quercetin, rutin), and pentacyclic triterpenoids (ursolic acid, hawthornic acid, oleanolic acid), which are also its main chemical constituents. This paper briefly reviews the chemical composition, nutritional value, food applications, and the important biological and pharmacological activities of hawthorn. This will contribute to the development of functional foods or nutraceuticals from hawthorn.

## 1. Introduction

As society develops and people’s living standards improve, various chronic diseases are emerging in the public. There has been a growing shift away from pharmaceuticals to wild medicinal plants, traditional herbs, and “ready-to-use healing foods” [1,2]. These traditional herbs and wild medicinal plants contain a large number of bioactive compounds. The properties of these plant ingredients allow them to be used in food and medicine. Among them, hawthorn of the Rosaceae family [3], known as the “nutritious fruit”, is attracting attention as one of the homologous medicinal herbs [4].

Hawthorn is a member of the Rosaceae family and is the most valuable species in the genus. It is often found on forest margins or in scrub on mountain slopes, mostly growing to 15 m. Its wood is hard and durable [5]. In the 2020 edition of the Pharma-copoeia of the People’s Republic of China (Part I), hawthorn is a wrinkled, uneven, round piece sliced, and dried in autumn after the fruit has ripened [6]. Hawthorn is widely distributed in the world, with over 1000 species [7], mainly in the temperate regions of the Northern Hemisphere. *C. monogyna* and *C. laevigata* are the main hawthorn species in Central Europe, *C. pentagyna*, *C. nigra*, and *C. azarolus* are the hawthorn species of southern and southeastern Europe. *C. pinnatifida* and *C. scabrifolia* are the main hawthorn varieties in China [8,9].

Hawthorn is widely used as a wild fruit for food and medical research. The fresh or dried fruits of hawthorn are used to make preserves, teas, and food supplements [10]. Extracts of hawthorn berries, leaves, and flowers are used to prevent hypertension and heart failure [11]. Hawthorn berries, leaves, and flowers are traditionally used in Europe for the treatment of heart disease, in North America for the same purpose [3,12], and in China, mainly for commercial products. In recent years, hawthorn is no longer restricted to the processing of some jams and snack foods. A growing number of functional hawthorn products are appearing on the market, such as hawthorn functional drinks, food additives, flavonoid injections, and hawthorn enzymes. Hawthorn can provide a variety of benefits to producers, food processors, and consumers and therefore has great economic value. However, due to the uniqueness of the product range and other reasons leading to some consumer limitations, there is an urgent need for further research and development of new hawthorn-based products.

This paper summarizes the nutritional and phytochemical composition, ethnomedical uses, food products and health benefits of hawthorn by reviewing web of science, PubMed, Google Scholar in conjunction with the scattered information available.

## 2. Nutrients and Phytochemical Compositions of Hawthorn

### 2.1. Nutrient Composition of Hawthorn

Approved as a medicinal fruit by the Chinese National Health and Wellness Commission [4], hawthorn has higher dietary fiber, pectin, ascorbic acid, minerals, and antioxidant capacity than some common fruits [13]. Studies have confirmed that hawthorn is rich in amino acids (8 essential amino acids and 3–8 times more amino acids than fruit), protein (17 times more protein than apple fruit), sugars, minerals (1st in calcium content among fruits), vitamins (vitamins A, C, B1, B2, about 10 times more vitamins), and has a high nutritional value [14]. Hawthorn is also rich in calcium, vitamin C, and carotene, with the highest calcium content and 890 mg/kg of vitamin, and rich in organic acids, which prevent vitamin C from being completely destroyed even when heated.

Hawthorn from different regions contains different levels of nutrients, depending on conditions such as variety, environment, genetics and plant harvest [4]. Hawthorn contains 7 mg/g of protein, 2 mg/g of fat, 30–40 mg/g of pectin, 30–50 mg/g of organic acids, 3–8 mg/g of tannins, 0.5–1.5 mg/g of amino acids, 0.89 mg/g of vitamin C, 0.89 mg/g of vitamin D, and 0.65 mg/g of flavonoids [15]. The crude protein, crude oil, ash, pH, acidity, and total phenolic content of native plants from central Anatolia, Turkey were 3.03%, 1.22%, 2.77%, 4.08, 1.56% and 9.35 mg/g, respectively [14]. A summary of nutritional composition of hawthorn in different regions can be seen in Table 1.

### 2.2. Phytochemical Compositions of Hawthorn

Hawthorn fruit, leaves, and flowers are rich in biologically active ingredients. It contains compounds with bioactive components such as organic acids, flavonoids, mucoxanthin, polyphenols, triterpenoids, and trace elements. Flavonoids range from 0.1 to 1.0% in the hawthorn fruit, 1 to 2% in the leaves and flowers, organic acids are second only to the flavonoids at 4.1% and proanthocyanidins range from 1 to 3% in the hawthorn fruit or leaves. Of these, total flavonoids and organic acids are the most abundant chemical constituents in hawthorn, while proanthocyanidins and total flavonoids are the two main categories of bioactive constituents in hawthorn. Hawthorn pericarp and pulp are also rich in pectin. Pectin extracted from hawthorn contains approximately 67% glyoxalate [16], has a high galacturonic acid and methyl esterification content and has a high viscosity compared to other food pectins such as lemon and apple pectin [17].

As a fruit rich in organic and phenolic acids, the organic acids in hawthorn are mainly malic, citric, succinic, ascorbic, quinic, oxalic, linolenic, and lauric acid [18]. The amount of organic acids varies depending on the variety of hawthorn. Citric and malic acids are the highest in hawthorn fruit, with malic acid averaging 1128.68 mg/100 g FW [18].

In addition, apart from organic acids, hawthorn is also rich in phenolic acids. The main phenolic acid in hawthorn is chlorogenic acid (8410–13826.7 μg/g), accounting for more than 80% of the total phenolic acid [19]. Gentisic acid, sinapinic acid, chlorogenic acid, quinic acid, protocatechuic acid, *p*-coumaric acid, *m*-coumaric acid, *o*-coumaric acid [20], caffeic acid, caffeic acid 3-glucoside [18], gallic acid, vanillic acid, syringic acid, ferulic acid, cinnamic acid [4], phlorodizin [21], neochlorogenic acid [22], salicylic acid, and ellagic acid [23]. Phenolic acids such as sinapic acid [24] have been quantified in previously studied hawthorn varieties. In leaves and flowers, none of the above acids have been measured more than once, so comparisons and generalizations of their concentrations cannot be made.

Flavonoids are the most abundant and wide-ranging class of compounds in hawthorn [25,26]. Currently, more than 60 flavonoids have been isolated from hawthorn [22,27,28,29,30,31,32,33,34], including the oxidized (chrysin, rutin, and quercetin) and reduced (proanthocyanidins, i.e., catechin derivatives) types [35]. According to its glycosides, it can be divided into flavonoids with apigenin, kaempferol, lignan, quercetin, and dihydroflavonoid glycosides as the main ones [36,37]. There are many varieties of hawthorn, and the total flavonoids vary from 2.27 to 17.40 mg/g in different varieties and different organs [35,38,39]. Table 2 lists the flavonoids that have been isolated from hawthorn.

Secondly, triterpenoids and their derivatives in hawthorn were isolated and identified in the first studies on hawthorn in the 1960s [40]. Triterpenoids are a wide range of structurally diverse compounds found in plants, consisting mainly of three terpene or isoprene units [41,42,43]. Hawthorn is a group of plants rich in triterpenoids, and some pharmacological studies have shown that these components have important effects such as cardiac strengthening, increasing coronary flow and improving circulation. And the presence of triterpenoids in the fruit may be the main reason for their anticancer activity [44]. Apart from the major triterpenoids such as ursolic acid [45] and oleanolic acid, all other terpenoids in hawthorn have been listed in Table 3.

Hawthorn seeds have been reported to contain high levels of lignan content [46]. Lignans are natural components synthesized by bimolecular polymerization of phenylpropanoids, which have antioxidant and anti-inflammatory activities and could be a new, inexpensive source of antioxidants, and inflammation inhibitors [47,48].

Sugars are mainly produced in the leaves and are transported to the fruit and other parts of the fruit as they develop. Sugar alcohols, mainly sorbitol and inositol, are mainly found in hawthorn varieties that are more commonly consumed as food [24]. Hawthorn homogeneous polysaccharides (HPS) are mainly a mixture of sugars including glucose, arabinose, caramel, galactose, galacturonic acid, rhamnose, and xylose. HPS contains two different molecular weight polysaccharide fractions of 1.423 × 10^5^ Da and 4.080 × 10^4^ Da respectively [49]. HAW1-2 is a polysaccharide with a molecular weight of 8.94 Da consisting of arabinose, galactose and glucose [50].

Hawthorn is rich in trace elements such as Ca, Fe, Mg, Cu, and Zn, with Ca content being the highest [4]. The content of trace elements varies slightly between different varieties of hawthorn. Chinese hawthorn berries contain 68 mg of sodium, 20 mg of phosphorus, and 2.10 mg of iron per 100 g of edible portion of hawthorn. The K, P, Ca, Mg, Fe, Na and B contents of hawthorn in the native plant of middle Anatolia in Turke were 16.3 mg/g, 1.3 mg/g, 1.3 mg/g, 0.9 mg/g, 0.06 mg/g, 0.06 mg/g, and 0.04 mg/g, respectively [51].

In addition to the above major bioactive components, hawthorn leaves also contain nitrogenous compounds. There are about 12 nitrogenous compounds isolated from hawthorn leaves, with components such as isobutylamine, ethylamine, dimethylamine, phenylethylamine, choline, acetylcholine, trimethylamine, isoamylamine, and ethanolamine [40]. Among them, phenylethylamine, tyramine, isobutylamine, and *o*-methoxyphenylethylamine are the main cardiac cardiac stimulant components. A total of 65 chemical components were identified in hawthorn essential oil, mainly eicosanoids (12–17%), twenty-one alkanes (11–16%), linalool (6–11%), hexadecanoic acid (1–11%), and nineteen alkanes (3–7%) [51,52].

To date, more than 170 compounds have been identified in hawthorn [53]. Figure 1 illustrates the chemical structures of several major bioactive components of hawthorn [12,19,35,44,53].

**Table 2 foods-11-02861-t002:** Flavonoids identified from Hawthorn.

Name	Species and Organ	Methodological and Analytical Approach	Extraction Solvent	References
Quercetin	*C. Almaatensis*; Flowers, fruits and leaves	HPLC-ESI-Q-TOF-MS and HRMS/MS	96% Ethanol or 50% ethanol	[20]
Quercitrin				
Cyanidin 3-glucoside				
Catechin				
Epigallocatechin				
Rutin				
Quercetin 3-glucoside (Isoquercetin)				
Vitexin 2″-*O*-rhamnoside				
Vitexin 4″-*O*-rhamnoside				
Vitexin				
Vitexin 4″-*O*-glucoside				
Quercetin glucoside				
Hyperoside	*C. Meyeri*; Fruits	HPLC	Methanol/water (80:20, 25 mL)	[12]
Epicatechin	NM	HPLC	80% Aqueous ethanol	[28]
Cyanidin chloride	*C. pinnatifida* var. *Major*; Fruits	HPLC-ESI-MS/MS	50% (*v*/*v*) Aqueous ethanol	[54]
Luteolin				
Apigenin				
Kaempferol				
Naringenin				
Phloretin				
Quercetin dirhamnosyl hexoside	*C. pinnatifida* Bge. var. *Major*; Fruits	HPLC-UV	80% Aqueous ethanol	[27]
Quercetin rhamnosyl hexoside				
Monoacetyl vitexin rhamnoside				
Luteolin-7-glycoside				
C-glycosides vitexin				
Sexangularetin-3-glucoside				
Sexangularetin-3-neohesperidoside				
Kaempferol-3-neohesperidoside				
Vitexin-2″-*O*-*α*-L-rhamnoside	*C. Oxyacantha*; Leaves and flowers	NM	Hydroalcoholic or water-based	[9]
Eriodictyol	*C. azarolus* Leaves	HPLC	Water/acetone mixture (1*v*/2*v*)	[55]
Hesperidin	*C. pinnatifida* Bge. or *C. pimiatificia* Bge. var. *major* N, E. Br; Fruits	UPLC/Q-TOF-MS	50% Ethanol	[56]
Rutoside	*C. oxyacantha* L; Fruits	HPLC, TLC, and UV	Hydroalcoholic or water-based	[26]
Isoquercitrin	*C. pinnatifida* Bge; Fruits	HPLC	80% Acetone	[38]
Hispertin	*C. azarolus* var. *eu-azarolus*; Leaves	RP-HPLC, UV, TLC, ^1^H NMR, and ^13^C NMR	Acetone, ethyl acetate, methanol and 70% ethanol	[23]
Chrysin				
Procyanidin trimers	*C. orientalis*, *C. szovitsii* and *C. tanacetifolia*; Leaves and twigs	VIP marker from OPLS-DA and UHPLC-ESI-QTOF-MS	Methanol	[37]
Isoxanthohumo				
Apigenin 7-*O*-glucoronide				
Chrysoeriol 7-*O*-(6″-malonyl-apiosyl-glucoside)				
Tetramethylscutellarein				
Myricetin 3-*O*-arabinoside				
Hydroxycaffeic acid				
*p*-Coumaric acid 4-*O*-glucoside				
Glycitin				
Quercetin 3-*O*-*β*-D-galactopyranoside	*C. Dahurica*; Fruits	HPLC, UV, ^1^H NMR, and ^13^C NMR	Methanol	[39]
Isorhamnetin 3-*O*-*α*-L-rhamnopyranosyl-7-*O*-*β*-D-glucopyranoside				
2″-*O*-rhamnoside	*C. pinnatifida* Bge; Leaves	HPLC-QTOF-MS	75% Ethanol	[53]
Orientin				
Iso-orientin				
Crataequinone A-B	*C. Pinnatifida*; Fruits	NM	NM	[24]
Pinnatifinosides A-D	*C. Pinnatifida*; Leaves			
Pinnatifins C-D,I				
1*β*, 9*α*-Dihydroxyeudesm-3-en-5β, 6α, 7α, 11*α* H-12, 6-olide	*C. Cuneata*; Fruits			
Proanthocyanidin A2	NM; Leaves and flower	HPLC	Acetone–water (7*v*/3*v*)	[29]
Proanthocyanidin B2				
Proanthocyanidin B4				
Proanthocyanidin B5				
Proanthocyanidin C1				
Proanthocyanidin D1				
Proanthocyanidin E1				
Epicatechin-(4*β*→6)-Epicatechin-(4*β*→8)-epicatechin				
Epicatechin-(4*β*→8)-epicatechin-(4*β*→6)-epicatechin				
Pinnatifinosides I	*C.* *pinnatifida* Bge. var. *major* N.E.Br; Leaves	UV, IR, MS, and 1D, 2D NMR	80% Ethanol	[30]
(+)-Taxifolin	*C. Sinaica*; Leaves	^1^H NMR and ^13^C NMR	70% Acetone	[31]
(+)-Taxifolin 3-*O*-xylopyranoside				
(+)-Taxifolin 3-*O*-arabinopyranoside 3-*O*-arabinopyranoside				
Crateside	*C. monogyna* and *C. pentagyna*; Leaves	UV	20% Ethanol	[32]
Neoschaftoside	*C. Monogyna*; Leaves	UV and TLC	Chloroform and butanol.	[33]
Neoisoschaftoside				
Cratenacin	*C. Curvisepala**;* Leaves	UV and IV	NM	[34]

NM: Not mentioned in the article; HPLC-ESI-MS/MS: high performance liquid chromatography-electrospray ion trap mass spectrometry; HPLC-QTOF-MS: high performance liquid chromatography-quadrupole-time of flight mass spectrometry; UPLC/Q-TOF-MS: ultra-high performance liquid chromatography-tandem quadrupole time of flight mass spectrometry; RP-HPLC: reverse phase high-performance liquid chromatography; VIP marker from OPLS-DA: VIP (variable importance in projection) selection method following supervised OPLS-DA; UHPLC-ESI-QTOF-MS: liquid chromatography coupled to quadrupole-time-of-flight mass spectrometry.

**Table 3 foods-11-02861-t003:** Terpenoids identified from Hawthorn.

Name	Species and Organ	Methodological and Analytical Approach	Extraction Solvent	References
Ursolic aldehyde	C. *dahurica*; Fruits	HPLC, ^1^H NMR and ^13^C NMR, UV	Methanol	[39]
Uvaol				
Ursolic acid				
Pomolic acid				
Euscaphic acid				
Tormentic acid				
3-Epi-2-oxopomolic acid				
2*α*, 19*α*-Dihydroxy-3-oxo-urs-12-en-28-oic acid				
Fupenzic acid				
2*α*-Hydroxy oleanolic acid				
Oleanolic acid				
Pinnatifidanoside A-D	*C. pinnatifida*; Leaves	HRESIMS, ^1^H NMR, and ^13^C NMR	75% Ethanol	[57]
Byzantionoside B				
(3*S*, 5*R*, 6*R*, 7*E*, 9*R*)-3,6-Epoxy-7-megastig men-5,9-diol-9-*O-β*-D-glucopyranoside				
(6*S*, 7*Z*, 9*R*)-Roseoside				
Icariside B6				
Linalool oxide *β*-D-glucoside				
Shanyenoside A				
Dihydrocharcone-2′-*β*-D-glucopyranoside				
Eriodectyol				
Shanyeside C,D,F	*C. pinnatifida*; Leave	HPLC-QTOF-MS	75% Ethanol	[53]
Euodionosides D				
Linarionoside A,B				
(6*S*, 7*E*, 9*R*)-6,9-Dihydroxy-4,7-megastiymadien-3-one-9-*O*-[*β*-D-xylopyranosy-(1→6)-*β*-D-glucopyranoside]				
Linalool oxide *β*-D-glucoside				
(6*R*, 9*R*)-3-Oxo-*α*-ionol-9-*O*-*β*-D-glucopyranoside				
Pisumionoside				
(3*S*, 5*R*, 6*R*, 7*E*, 9*S*)-Megastiman-7-ene-3,5,6,9-tetrol				
Pinnatifidanoside G				
Norhawthornoid B				
Corosolic acid	*C.**P**innatifida*; Fruits	HPLC	80% Acetone	[44]
Maslinic acid				
(3*R*,5*S*,6*S*,7*E*,9*S*)-Megastigman-7-ene-3,5,6,9-tetrol 9-*O*-*β*-D-glucopyranoside	*C. Pinnatifida*; Leaves	^1^H NMR, ^13^C NMR, HSQC, HMBC, and NOESY	70% Ethanol	[42]
(6*S*,7*E*,9*R*)-6,9-Dihydroxy-4,7-megastigmadien-3-one 9-*O*-[*β*-D-xylopyranosyl-(1″→6′)-*β*-D-glucopyranoside]				
Linarionoside A-C	*C. Pinnatifida*; Leaves	^1^H NMR and ^13^C NMR	70% Ethanol	[43]
3*β*-D-Glucopyranosyloxy-*β*-ionone				
Icariside B_6_				
Pisumionoside				
(3*S*,5*R*,6*R*,7*E*,9*R*)-3,6-Epoxy-7-megastigmen-5,9-diol-9-O-*β*-Dglucopyranoside				
(6*S*,7*E*,9*R*)-Roseoside				
(6*R*,9*R*)-3-Oxo-α-ionol-9-*O*-*β*-D-glucopyranoside				
4-[4*β*-*O*-*β*-D-Xylopyranosyl-(1″→6′)-*β*-D-glucopyranosyl-2,6,6- trimethyl-1-cyclohexen-1-yl]-butan-2-one	*C. P**innatifida*; Leaves	^1^H NMR, ^13^C NMR, HSQC, HMBC, and NOESY	70% Ethanol	[42]
(3*S*,9*R*)-3,9-Dihydroxy-megastigman-5-ene 3-*O*-primeveroside				
(3*R*,5*S*,6*S*,7*E*,9*S*)-Megastiman-7-ene-3,5,6,9-tetrol				
(5*Z*)-6-[5-(2-Hydroxypropan-2-yl)-2-methyltetrahydrofuran-2-yl] -3-methylhexa-1,5-dien-3-ol				
(5*Z*)-6-[5-(2-*O*-*β*-*D*-Glucopyranosyl-propan-2-yl)-2-methyl tetrahydrofur-an-2-yl]-3-methylhexa-1,5-dien-3-ol				
5-Ethenyl-2-[2-*O*-*β*-D-glucopyranosyl-(1′′→6′)-*β*-*D*-glucopyranosyl-propan-2-yl]-5-methyltetrahydrofuran-2-ol				

HPLC-QTOF-MS: high performance liquid chromatography-quadrupole-time of flight mass spectrometry; HRESIMS: high resolution electrospray ionization mass spectroscopy.

## 3. Applications in Food Products

Fresh hawthorn fruit can be eaten directly without any fumigation or washing. To meet the high quality standards expected by consumers, hawthorn fruit has been processed into many types of products. With the continued development of science and technology, such as the enzyme industry [58], homogenizer products and membrane technology, a new phase in the processing of hawthorn has been introduced and its product range is becoming more and more diverse. Figure 2 shows modern food applications of hawthorn.

### 3.1. Traditional Hawthorn Products

Hawthorn is rich in nutritional value, stimulating appetite and digestion, and is widely used in the food industry [15]. There are many products made from hawthorn on the market, with more than 150 types of products sold. Traditional hawthorn products in China mainly include sugar gourd, hawthorn cakes, hawthorn preserves, canned hawthorn, hawthorn chips, and hawthorn roll.

### 3.2. Bakery Products

In the production of hawthorn bakery products, it is most important to use effective methods to retain its effective active ingredients and increase the absorption and dissolution properties of hawthorn in bakery products. Wang et al. [59] used ultra-fine grinding technology to produce hawthorn bread with 3% hawthorn powder addition, 0.6% salt, 18% sugar, and 0.5% bread amendment as a formula to improve its health benefits and special flavor.

The addition of hawthorn to the whole wheat flour bread has the effect of promoting the normal function of the digestive and circulatory system and also has an anti-hyperglycemia effect, this raw material is inexpensive and can be consumed by people with type 2 diabetes [60].

### 3.3. Brewing Products

Hawthorn is so rich in vitamins, minerals, and active substances that can meet people’s nutritional requirements for fruit wines. The sales of fruit wines in China are increasing at a rate of 15% and the development prospects are very promising. Beers produced from the North American shrub, star-spotted hawthorn, not only have a higher antioxidant capacity and higher polyphenol concentration, but also have a greater degree of improvement in taste, aroma, clarity, and overall impression [61]. Treatment of hawthorn wine with pectinase facilitated the reduction of pH and the release of methanol in the wine and accelerated the clarification of hawthorn wine [62].

With the widespread use of hawthorn, vinegar has become more than a simple flavoring agent, and its health maintenance and wellness benefits are receiving more and more attention. The production of hawthorn vinegar from hawthorn berries not only has good characteristics of volatile aromatic compounds that enrich the taste of the vinegar, but also has highly bioactive phenolic compounds that exert a nutritional and health value, increasing the area of use and consumption of hawthorn berries [63]. The use of hawthorn for the production of hawthorn vinegar is a more efficient way to produce innovative and healthy products.

### 3.4. Beverages

Pectin extracted from hawthorn wine residue was used to produce yogurt and was able to improve the stability and sensory acceptability of the yogurt, which offered the possibility of developing hawthorn wine by-products [64]. Microencapsulated *Lactobacillus rhamnosus* GG in hawthorn berry tea met the minimum requirement of 106–107 cfu/mL for probiotics, which can play a therapeutic and health care role, and the optimized microspheres of microencapsulated *Lactobacillus rhamnosus* GG could improve the functional properties of hawthorn tea. *Lactobacillus rhamnosus* GG tea would be a functional food product with potential added value that could be marketed [65]. Water kefir beverages (lactic acid bacteria drinks) prepared with hawthorn as a food matrix have a reduced sugar content during the fermentation process when fermented sugars are converted to ethanol and carbon dioxide. This is an important alternative for consumers who are unable to consume dairy products as a source of beneficial microorganisms [66].

Hawthorn drinks have unlimited potential both in terms of taste and nutritional value, but the acid content of hawthorn drinks is high. The removal of organic acids from hawthorn improves the organoleptic properties but results in the loss of the important functional compound TF (flavonoids) in the deacidification process. In China, there is currently no hawthorn juice with acceptable acidity and functional ingredients also present in the original juice [67].

### 3.5. Meat Products

The active ingredients in hawthorn have anti-cancer and anti-oxidant properties. Using a recipe of 6% jam, 6.5% rose paste, 5% milk powder and 0.004% sodium nitrite, Zhou et al. [68] incorporated hawthorn and rose into traditional sausages, giving them a new taste and flavor, and solving the food safety problem of nitrite in traditional cured sausages. The antioxidant and antibacterial effects of phenolics in hawthorn have also been investigated in a variety of commercial foods such as lamb burgers, frankfurters, and pork liver [12].

### 3.6. Jams

As the perfect companion to bread and other pastries, the consumer demand for jam is gradually increasing. According to China Industry Research Network, the consumption of jam in China has been growing at a rate of around 15%. Hawthorn jams on the market at present are broadly divided into two types: an ordinary hawthorn jam and compound jam with hawthorn flavor, such as hawthorn leaf flavonoid jam [69] and hawthorn passion fruit jam [70].

Li et al. [59] used 1.15% hawthorn leaf flavonoids, 45% white sugar, 0.30% pectin, 0.20% xylitol, and 0.16% citric acid as the formula to develop hawthorn leaf flavonoid jam with flavonoid flavor. The composite hawthorn jam not only meets the need for a safe and healthy jam, but also further enhances the use of hawthorn and hawthorn by-products.

### 3.7. Sugar Products

Hawthorn candy products can change the sour and astringent taste of hawthorn itself and expand the consumer market for hawthorn. Combining hawthorn with fondant not only enriches its taste, but also adds nutritional health functions to fondant [71].

With the increasing demand for hawthorn products, China produces approximately 1.5 million tons of hawthorn kernels (HK) annually. Hawthorn kernels, as a major by-product in the hawthorn processing industry, account for approximately 30% of the hawthorn, and as an agricultural lignocellulosic biomass containing 28% xylan, it is valuable to use HK to develop value-added products such as xylose and oligosaccharides (XOS) [72].

## 4. Ethnomedicinal and Biotechnological Uses

The most common traditional use of hawthorn is as a food, whether eaten as a fruit or in hawthorn products, which are very popular. Hawthorn has ethnomedicinal value for its digestive and anti-cardiovascular properties. In Europe, the use of hawthorn as an herbal remedy dates back to the late 19th century. The use of its thorns as a dermatological aid to detect boils and treat swellings (i.e., rheumatism), Kwa-kiutl, Okanagon and Okanagan-Colville and the use of decoctions by the Iroquois as a witchcraft medicine to cause or prevent a person from ‘breaking out like a cancer’ [24]. Hawthorn also contains bioactive components with great potential for the pharmaceutical industry, including polyphenols and flavonoids.

Hawthorn berries are rich in flavonoids, which have been used as reducing and stabilizing agents in the green synthesis of CP-AuNps and CP-AgClNps nanoparticles as antimicrobial agents [73]. Hawthorn kernels, a by-product of approximately 30% of hawthorn berries, contain a large amount of agricultural lignocellulosic biomass of 28% xylan, which has the potential to be processed into xylose [72]. In addition, the many active plant polyphenols in hawthorn, which have a delaying effect on browning, are a very promising alternative to anti-browning substitutes [74].

## 5. Health Benefits

The active ingredients in hawthorn are the basis for the wide range of health benefits it exerts (Table 4). For example, polyphenolic compounds are good antioxidants [12] and immunomodulators [75], flavonoids have anti-inflammatory [76] and anti-atherogenic activity [77], lignans have antibacterial [78] and antioxidant activity [39], while triterpenoidshave anticancer [79], anti-inflammatory, and anti-proliferative activities [41]. It is therefore essential to advocate the exploitation of the therapeutic potential of hawthorn in new food products to improve and maintain the nutritional and health indices of the population.

### 5.1. Anticancer

At present, there is little information and research on hawthorn’s anticancer effects. However, the bioactive substances contained in hawthorn are believed to have beneficial effects on human cancer cells [62]. The anticancer effects of hawthorn can be divided into two main areas: hawthorn extract and hawthorn isolated compounds.

Qiao et al. [17] found that triterpenoids extracted from hawthorn with acetone had significant inhibitory effects on the proliferation of human breast and hepatocellular carcinoma cells. Ursolic acid, a triterpenoid, has been shown to induce apoptosis of MDA-MB-231 cancer cells via the mitochondrial pathway [44]. (−)-Epicatechin (EC) was extracted from the total oligomeric flavonoid (TOF) extract of hawthorn. At the same incubation time (48 h), 400 μg/mL of TOF and 200 μg/mL of EC could reduce the viability of melanoma cells in B16F10 mice by 77.57% and 66.55%, respectively. Meanwhile, the mean tumor weight (0.24 g) and volume (242 mm^3^) of the mice in the TOF extract group were significantly lower than those in the control group (1.5 g, 3653.75 mm^3^), indicating that TOF extract could exert anti-tumor effects by inhibiting the growth of tumors in vivo [55].

Similarly, compounds isolated from hawthorn have a wide range of anticancer activities. The apoptosis rate of human neuroblastoma SHSY-5Y increased from 9 to 54% with increasing doses of hawthorn acid. Maslinic acid inhibited the activity of anti-apoptotic proteins Bcl-2 and Bcl-xL in SHSY-5Y cells, and the activity of pro-apoptotic Bax protein was significantly enhanced, which exerted anti-proliferative effect and achieved anticancer purpose [79]. In addition, hawthorn polysaccharide can inhibit the proliferation of human colon cancer cells. 1000 μg/mL of Hawthorn Polysaccharide (HPS) could significantly increase the proportion of G2/M phase (17.35%) and S phase (67.82%) of tumor cells. Also, it increases the rate of apoptosis of cancer cells [49]. Hawthorn isolated compounds have shown anti-tumor activity both in vivo and in vitro and are considered a promising drug for the treatment of melanoma [55].

### 5.2. Cardiovascular System

The proanthocyanidins, total flavonoids, and other extracts contained in the fruits, leaves, and flowers of hawthorn are commonly used in the treatment of cardiovascular disease due to their significant in vitro effects and apparent safety profile [25,80,81,82,83,84,85,86] (Figure 3).

#### 5.2.1. Anti-Hypertensive

The diastole of the peripheral blood vessels is the main cause of the fall in blood pressure [87]. Hawthorn fruit extract confers antioxidant effect on high-salt induced hypertension and it may be used as a nutritional supplemental therapeutic drug to protect against high-salt induced hypertension in renal medulla [88]. A randomized controlled trial of type 2 diabetes showed that patients given hawthorn extract (1200 mg per day) had a significantly greater reduction in mean diastolic blood pressure than those given placebo (the prescribed drug) [9]. The hydroalcoholic extract of hawthorn flower heads was shown to inhibit thromboxane A2 (TXA2) activity and promote peripheral vasodilation, thereby lowering blood pressure [89].

#### 5.2.2. Lipid Regulation and Anti-Atherosclerosis

Non-alcoholic fatty liver and obesity caused by disorders of lipid metabolism have become serious social and health problems worldwide [90]. Hawthorn extract has the ability to regulate blood lipid levels and anti-atherosclerotic effects [82,91].

Hawthorn treated with n-ethanol and ethyl acetate alleviated hyperlipidemia by regulating disorders of lipid, energy and amino acid metabolism and reducing oxidative stress, resulting in a decrease in plasma levels of total cholesterol (TC), triglycerides (TG), and low density lipoprotein cholesterol (LDL-C) in hyperlipidemic rats [92]. Similarly, Hawthorn extract was able to improve hyperlipidemia by improving the lipid profile, reducing oxidative stress and lowering total and LDL cholesterol levels in the serum of ovariectomized rats (OVX) [83]. In addition, the alcoholic extract of hawthorn is able to prevent atherosclerosis. The alcoholic extract of Zhongtian hawthorn promotes cholesterol elimination [93]. Mice with knockout apoE gene showed a significant reduction of 23.1% in the area of atherosclerotic lesions by feeding hawthorn leaf flavonoids [76].

#### 5.2.3. Cardioprotective Effect

WS 1442, a dried extract of Hawthorn with flowering leaves (4–6.6:1), has been shown to have positive inotropic and anti-arrhythmic properties [94]. Hawthorn ethanol extract (HAE) can increase the activity of glutathione peroxidase (GSH-Px) and decrease the content of malonaldehyde (MDA) in the myocardium, alleviating inflammatory cell infiltration, and myofibrillar disorders in the myocardial structure [84]. Hawthorn leaf flavonoids (HLF) protect against diabetes-induced cardiomyopathy by reducing oxidative stress and inflammation through the PKC-α signaling pathway [25]. The citric acid, caffeic acid, chlorogenic acid, and quercetin in hawthorn also have a protective effect against myocardial ischemia [82]. Dried hawthorn berries treated with honey can increase the content of organic acids, phenylpropanoids, and flavonoids in hawthorn, thus further enhancing the anti-myocardial ischemia effect of hawthorn [22].

### 5.3. Anti-Hyperglycemic

Diabetes mellitus is a chronic metabolic disease characterized by high blood sugar [85]. The phenolic and flavonoid compounds in hawthorn are considered to be important active substances in the fight against diabetes. Hawthorn is able to improve hyperlipidemia by lowering blood sugar and triglyceride levels caused by a high-fat diet [85]. Human *α*-glucosidase is the main enzyme catalyzing the final step of carbohydrate hydrolysis in the digestive system [86,95]. The quercetin in hawthorn can effectively inhibit the activity of alpha-glucosidase enzyme, resulting in a reduction in the release and absorption of glucose thereby lowering blood sugar levels [96,97]. In addition, hawthorn leaf flavonoids have a protective effect on the myocardium of streptozotocin-induced diabetic rats [25], and the mechanism may be due to the reduction of oxidative stress and inflammation through the PKC inactivation pathwayction in the release and absorption of glucose thereby lowering blood sugar levels [25].

### 5.4. Antibacterial and Anti-Inflammatory

The lignans, polysaccharides and total flavonoids in hawthorn exert their anti-inflammatory effects mainly by inhibiting the activity of inflammatory factors such as NO and TNF-*α*. Among the compounds isolated by Peng et al., lignans **7** to **12** showed strong inhibition of NO, especially compound **12** (IC_50_: 50.5 μM) showed stronger NO inhibitory activity than dimethyltetracycline (IC_50_: 55.1 μM). While compounds **1**–**4** had inhibitory effects on TNF-*α* with IC_50_ values of 76.1, 47.9, 84.4, and 94.2 μM respectively [48].

Hawthorn pectin (POS) improves liver inflammation by reducing total liver fat content and levels of TNF-*α* and IL-6, increasing IL-10 levels and inhibiting NF-*κ*B activation [98]. HAW1-2 suppresses colitis by inhibiting inflammatory factors such as TNF-*α*, modifying the gut microbiota and producing SCFAs [50]. Similarly, hawthorn aqueous extract significantly reduced LPS-induced NO production in RAW cells to protect RAW264.7 cells from lipopolysaccharide (LPS) induced apoptosis after 24 h stimulation, resulting in 45.7% of cells surviving [99].

Hawthorn seed extract showed antibacterial activity against vaginitis pathogens (*Escherichia coli*, *Staphylococcus aureus*, *Pseudomonas aeruginosa*, *Gardnerella vaginalis*, *Candida albicans*), especially 2,6-dimethoxyphenol in hawthorn seeds could inhibit the growth of *vaginitis pathogens* [67].

Hawthorne acid (MA) can also be used for arthritis, especially in the elderly where arthritis is now very common. The pain caused by arthritis affects the patient’s normal life and although not life-threatening, it causes great discomfort to the patient. Although MA has recently been found to relieve arthritis pain, its anti-inflammatory mechanism still needs to be further explored and studied to lay a solid foundation for its clinical application [100].

### 5.5. Antioxidant

Hawthorn has hypolipidemic and vascular protective effect, which is mainly attributed to its high antioxidant effect. Phenolic compounds, flavonoids, and triterpenoids are important components of natural antioxidant substances.

In hawthorn fruits, free phenolic compounds accounted for 35.3–37.8% of the antioxidant activity, followed by insoluble bound phenolic compounds accounting for 25.0–27.0% of the antioxidant activity [101]. Hawthorn polyphenol extract (HBE) can regulate the apoptosis of HaCaT cells by reducing reactive oxygen species (ROS) production, DNA damage and p53 activation, effectively reducing UVB-induced photodynamic and non-photodynamic damage and providing protection to the skin [54,102].

Hawthorn total flavonoid extract (TOF) and (−)-epicatechin (EC) inhibited acute cellular oxidative stress triggered by the free radical initiator ABPA. And both TOF extract and EC at all concentrations tested were taken up by cells and both significantly inhibited the oxidation of DCFH [55].

### 5.6. Anti-Digestion

Hawthorn is a common herbal remedy for indigestion, and its herbal tonics and extracts have been shown to improve gastrointestinal motility. The phenolic substances in hawthorn, especially phenolic acids, play an important role in the digestive system due to their stable structure [103].

Hawthorn soup can relieve indigestion caused by high-calorie diet (HC-DID). The bidirectional action of brain and intestine is the main mechanism for treating HC-DID. Gavage administration affects the “brain-intestine” axis and gavage administration affects the “intestine-brain” axis. Hawthorn decoction with hawthorn charcoal can improve the brain-gut polypeptide disorder caused by HC-DID by regulating the “brain-gut” axis, significantly reducing the ratio of bacterial gate and mycelium, and improving the intestinal metabolic disorder [91].

Hawthorn extract protects the gastrointestinal tract mainly by altering disorders of lipid, energy, and amino acid metabolism. In the atropine gastrointestinal motility disorder model, the ethanolic extract of hawthorn modulates the metabolism of bile acids and promotes the excretion of lipids to some extent. At the same time, the synthesis of CoA increased, the content of the intermediate acetyl-CoA also increased, and the content of L-tryptophan and serotonin increased. This indicates that the metabolism of the three major nutrients in the model rats was restored to normal and the function of the gastrointestinal tract was improved [92]. Hawthorn seed ethyl acetate extract (HAESE) can lower blood glucose, prevent weight loss, accelerate gastric emptying and small intestine propulsion, effectively regulate plasma gastrointestinal hormones such as gastric hunger hormone, motilin (MTL), and gastrin (GAS), promote gastrointestinal motility in rats with diabetic gastroparesis and alleviate the symptoms of diabetic gastroparesis (DGP) rats [57,104].

### 5.7. Others

#### 5.7.1. Immune Regulation

Gluten sterols isolated from hawthorn significantly increase white blood cell counts and enhance phagocytic activity of macrophages [40]. Hawthorn phenolic extracts have a strong stimulatory effect on splenocytes and lymphocyte subsets. This is mainly due to the fact that epicatechin and proanthocyanidins B2 and B4 in hawthorn promote the proliferation of splenocytes, significantly increase the percentage of B cells, decrease the percentage of Th cells and stimulate the humoral immune response. And the effect is more pronounced at high doses (100 and 200 mg/kg) of the extract [75].

#### 5.7.2. Anticoagulant

Components of hawthorn such as bioflavonoids and proanthocyanidins have beneficial effects on the blood coagulation system. In Sprague Dawley rats, continuous administration of hawthorn extract revealed a significant increase in cardiac antithrombin III and a significant decrease in their hepatic factor X, suggesting that hawthorn has blood thinning properties [105]. The polyphenol-polysaccharide conjugate isolated from the hawthorn fruit prolonged the coagulation process of plasma in vitro experiments, even at concentrations as low as 31.25 μg/mL. However, only the product isolated and processed from the floral source was highly selective, mainly because it was a non-direct inhibitor of the non-thrombin-mediated factor Xa [106]. Therefore, special attention should be paid to the use of anticoagulant/antiplatelet agents when using hawthorn extract.

#### 5.7.3. Neuroprotective

Alzheimer’s disease (AD) is one of the major neurodegenerative diseases in which the main component is beta amyloid (A*β*) and oligomers of A*β* have direct neurotoxicity mainly through membrane receptors, channels and intracellular dysfunction [107].

Hawthorn ethanol extract (CPE) has long been used as a traditional medicine for the treatment of various diseases, and it mainly blocks the aggregation of A*β* and degrades A*β* protofibrils in a concentration-dependent manner [108]. Among the seven lignans isolated from hawthorn seeds, compounds 5 and 6 showed strong inhibition of A*β*1-42 and were able to inhibit the self-induced A*β* aggregation activity [46].

Similarly, depression, the most common chronic mental disorder, has a huge impact on society. Flavonoid and proanthocyanidin components of hawthorn, such as chlorogenic acid, inhibit stress hormone-induced depressive behavior via hippocampal astrocyte monoamine oxidase B-reactive oxygen species signaling [109].

Total flavonoids from hawthorn leaves can also reduce apoptosis in rats with spinal cord injury, exert neuroprotective effects and promote the recovery of motor function in rats with spinal cord injury [110].

**Table 4 foods-11-02861-t004:** Biological activity and mechanism of action of Hawthorn.

	Extracts or Compounds	Observation or Methods	Effects	References
Anticancer activity				
	Triterpenoids isolated from hawthorn berries	In vitro, MTT assay.	All 15 triterpenoids showed effective antiproliferative activity against human HepG2, MCF-7 and MDA-MB- 231 tumor cells showed potent anti-proliferative activity(compound **2**–**4** EC_50_ < 5 µM).	[111]
	Ursolic acid, oleanolic acid, corosolic acid, and maslinic acid	In vitro.	CA showed the highest antiproliferative activity against human HepG2 (EC_50_ = 9.44 μM), MCF-7 (EC_5__0_ = 22.01 µM) and MDA-MB-231 (EC_50_ =26.83 μM) tumor cells among the four triterpenoids, followed by UA, MA, and OA.	[44]
	Phenylpropanoids isolated from hawthorn fruit	In vitro, MTT assay.	Five compounds (**1a**/**1b**, **2**–**4**) were used in the treatment of human HepG2 and Hep3B cells with better cytotoxicity(**1a** IC_50_: 59.57, >100 µM; **1b** IC_50_: 35.37, 70.42 µM; **2** IC_50_: 27.36, 39.40 µM; **3** IC_50_: 18.68, 38.96 µM;**4** IC_50_: 17.50, 43.58 µM;).	[112]
	Homogeneous polysaccharide (HPS)	In vitro, WST-1 colorimetric method.	Treatments with 500 and 1000 μg/mL of HPS for 12 h resulted in more than 74% of growth inhibition against human HCT116 cell.	[49]
	An extract enriched with TOF	In vivo.	TOF extract from hawthorn leaves exerts an antitumor effect by decreasing the melanoma tumor growth in vivo (6 times less weight).	[55]
	Maslinic acid	In vitro, AO/EB staining assay and Annexin V/PI dual staining.	It can cause human neuroblastoma SHSY-5Y cells increase the percentage of apoptotic cells from 9% in the control group to 54% at higher drug doses.	[18]
Cardiovascular system activities				
	Hawthorn Leaf Flavonoids(HLF)	In vivo.	HLF protect against diabetes-induced cardiomyopathy in rats via PKC-*α* signaling pathway.	[25]
	Hawthorn Leafs Extract	In vitro, MTS.	Hydroalcoholic extracts of hawthorn leaves at 300 and 1000 mg/mL significantly reduced the frequency of arrhythmias induced by adrenaline stimulation.	[82]
	Hawthorn Fruit Extract (HFE)	In vivo.	HFE could dose-dependently reduce the TMAO-aggravated atherosclerosis.	[113]
	Flavonoids	In vivo, carrageenan-induced tail thrombosis model.	Inhibiting TXA2 release, decreasing the level of Ca^2+^ in platelets or blocking glycoprotein IIb/IIIa receptors may be the mechanism of the antithrombotic effects of flavonoids.	[81]
	Hawthorn Fruit Extract	In vivo, Western Blot.	The hepatic triglyceride (TG) and malondialdehyde (MDA) levels were significantly reduced in the hawthorn groups compared with the ovariectomized group (*p* < 0.05).	[83]
	Hawthorn Fruit Extract	In vivo, spectrophotometry.	Compared with the blood TC levels of rats in the type 2 diabetic group, the blood TC levels of rats in the high, medium and low dose of Hawthorn extract decreased by 162.54%, 122.68% and 92.13% respectively.	[86]
	Hawthorn Fruit Extract	In vivo.	Echocardiographic parameters (LVESD, LVEDD) were reduced in rats with chronic heart failure treated with hawthorn extract (*p* < 0.01)	[84]
	Hawthorn Extract	In vivo.	Hawthorn extract groups suppressed the high-fat diet-induced increases in the concentrations of LDL (*p* < 0.05).	[85]
Anti-hyperglycemic activity				
	Hawthorn Fruit Extract	In vivo.	Hawthorn extract in high, middle and low dose could significantly reduce the fasting blood glucose levels of type II diabetic rats from 20.25 ± 1.9 mmol L^−1^ to 10.5 ± 0.87 mmol L ^−1^, 15.13 ± 0.55 mmol L ^−1^ and 17.9 ± 0.87 mmol L^−1^ (*p* < 0.01 and *p* < 0.05).	[86]
	Hawthorn polyphenols, D- chiro- inositol (DCI), and epigallocatechin gallate (EGCG)	In vitro.	Three ingredients exerted the synergistic hypoglycemic effect to enhance glucose consumption and glycogen levels and inhibit hepatic gluconeogenesis in IR-HepG2 cells.	[95]
	Hawthorn Extract	In vivo.	Hawthorn treated groups (0.5 g/kg/day, 1.0 g/kg/day) showed a significant reduction in insulin resistance compared with the HF group (*p* < 0.05, *p* < 0.01).	[85]
Antibacterial and anti-inflammatory activities				
	Hawthorn Fruit Extract	In vivo.	The hawthorn treatment group reduced the levels of IL-6, IL-8, IL-1*β* and TNF-*α* in cardiomyocytes due to doxorubicin treatment for heart failure (*p* < 0.01).	[93]
	Water fraction from hawthorn fruit	In vitro, ELISA.	Water fraction from hawthorn fruit at 200, 400 and 600 µg/mL increased the survival rate of RAW264.7 cells to 61.8%, 72.7% and 83.4% respectively.	[99]
	Hawthorn Methanolic Extract (ME)	In vitro.	ME from hawthorn had a minimum MIC and MBC value of 1.25 µg/mL against *S. aureus* and *S. typhimurium*.	[38]
	Hawthorn polysaccharide (HAW1-2)	In vivo.	The relative expression of IL-1*β*, IL-6 and TNF-*α* were suppressed after HAW1–2 treatment.	[50]
	Hawthorn phenolic extract	In vivo.	The extract decreased the percent-age of CD^4^^−^CD^8^^−^ and CD^4+^ thymocytes but elevated the percentage of CD^4+^CD^8+^ and CD^8+^ thymic cells, increased the total number, percentage, and absolute count of T and B splenocytes.	[75]
	Pectin oligosaccharide (POS)	In vivo, ELISA.	Higher dose (0.75, 1.5 g/kg) of POS significantly (*p* < 0.01) decreased the contents of hepatic TNF-*α* and IL-6, while significantly (*p* < 0.05–0.01) increased the level of IL-10, compared with the high fat control group.	[98]
	Total Flavonoid Extract from Hawthorn (TFH)	In vitro.	TFH (50–200 µg/mL) treatment inhibited the increase of inflammatory cytokines IL-6, IL-1*β*, MCP-1 and IL-8 in Caco-2 cells in a dose-dependent manner.	[76]
Anti-digestion activty				
	Hawthorn Seed Eextract (HSEAE)	In vivo. ELISA.	Different doses of HSEAE effectively promoted the gastric emptying and small intestinal propulsion (*p* < 0.05 or *p* < 0.01). In addition, HSEAE increased SOD and GSH-Px in the rats’ stomachs while decreasing MDA, and increased plasma ghrelin while decreasing MTL and GAS (*p* < 0.05 or *p* < 0.01).	[104]
	Ethyl acetate part of hawthorn	In vivo, LC-MS.	The effect of ethyl acetate extract of hawthorn on gastric emptying rate and intestinal propulsion rate in a rat model of atropine sulfate-induced gastrointestinal motility retardation was significant (*p* < 0.05, *p* < 0.001).	[92]
	Charred hawthorn	In vivo.	Hawthorn decoction coupled with the odor of charred hawthorn effectively alleviate high-calorie-diet-induced dys-pepsia in rats by regulating the “Brain-Gut” axis and gut flora.	[91]
Antioxidant activity				
	Triterpenoids isolated from hawthorn berries	In vitro, PSC and superoxide anion free radical assay.	In PSC assay, compounds **1**, **10** and **12** had pronounced antioxidant activity with an EC_50_ of 0.2 ± 0.01, 0.5 ± 0.01, and 0.7 ± 0.01 µM.	[111]
	Phenolic composition of Kazakh Crataegus	LC-MS	In the free radical scavenging activity assay (DPPH), the most potent extract was the phenolic compound from hawthorn leaves (IC_50_ 48 ± 2 µg/mL).	[20]
	Hawthorn polyphenol extract (HPE)	In vivo and vitro, MTT.	After UVB irradiation, the cell viability significantly decreased (*p* < 0.05). HPE at 5 and 10 µg/mL significantly increased cell survival (*p* < 0.05).	[102]
	Phenolic compounds	In vitro, ORAC.	The antioxidant activity of phenolic compounds in hawthorn was significant, with ORAC values for the eight phenolic compounds ranging from 5.25 ± 0.54–62.79 ± 1.46 μmol TE/μmol.	[114]
	Hawthorn fruit extract	FRAP.	The antioxidant activity was widely varied (*p* < 0.001) in species of *Crataegus*, ranging from 0.32–1.84 mmol Fe^++^/g DW.	[12]
	Phenolic compounds	DPPH, ABTS, and FRAP.	The total antioxidant activity of organic fresh hawthorn berry fruit determined by DPPH, FRAP and ABTS assay was up to 286 ± 4, 320 ± 5 and 328 ± 6 μmol TE/g DW.	[54]
	Hawthorn extract	DPPH.	The DPPH scavenging capacity of the fresh hawthorn slices was 3.48 mmol TE/100 g DW.	[101]
	Extract from peel of hawthorn fruit(EPHF)	DPPH and ORAC.	EPHF has the strongest oxygen radical scavenging capacity (IC_50_ = 11.72 μg/mL).	[115]
	Organic freeze-dried hawthorn berries (OFDHB)	ABTS, FRAP and DPPH.	The peel of OFDHB sample had the highest antioxidant capacity followed the decreasing order of ABTS (577.5 µmol TE g^−1^) > FRAP (455.84 µmol TE g^−1^) > DPPH (410.75 µmol TE g^−1^) assay.	[4]
	Flavonoids	FRAP.	The highest antioxidant activity was observed in the leaves of *C. pentagyna* as 4.65 mmol Fe^++^/g DW, whereas the lowest activity (0.9 mmol Fe^++^/g DW) was found in the leaves of *C. azarolus* var. *aronia*.	[116]

PSC: peroxyl radical scavenging capacity; MTS: 3-(4,5-dimethylthiazol-2-yl)-5-(3-carboxy-methoxyphenyl)-2-(4-sulfophenyl)-2H-tetrazolium assay; ELISA: enzyme-linked immunosorbent assay; ORAC: oxygenradical absorbance capacity; DPPH: 1,1-diphenyl-2-picrylhydrazyl; FRAP: ferric-reducing antioxidant Power; ABTS: Total Antioxidant Capacity Assay Kit with ABTS method; **1a**/**1b**, **2**–**4**: (+)-crataegusanoid A, (−)-crataegusanoid A, crataegusanoid B, crataegusanoid C, crataegusanoid D.

## 6. Conclusions

Hawthorn is a good source of proteins, amino acids, sugars, minerals, vitamins and phytochemicals including terpenoids, phenols, and flavonoids. As a medical herb, hawthorn contains a large number of naturally occurring bioactive chemicals with therapeutic properties and is therefore a highly marketable source of medicines worldwide. However, further in vivo and in vitro studies and clinical trials are needed to assess the link between the chemical composition of hawthorn and the mechanisms of action for the treatment of various diseases.

Dozens of bioactive substances in hawthorn can be extracted as a health product. At present, the processing technology of hawthorn food has been widely emphasized, both traditional processing technology and modern technology. Hawthorn has been processed into many products, such as hawthorn drinks, hawthorn paste, hawthorn vinegar, hawthorn wine and bread. Hawthorn kernels have even been used to produce value-added products such as xylose. However, due to the excessive organic acid content in hawthorn, a large amount of sugar needs to be added for flavouring when producing hawthorn products, which prevents many elderly and diabetic patients from enjoying the health benefits of hawthorn. With the research of deep processing technology of hawthorn, the proper ratio of hawthorn with other materials, the development of new products, the improvement of traditional processing methods, minimizing the loss of functional factors and natural pigments of hawthorn, and the development of new areas of hawthorn utilization, such as: the utilization in special medical use formulae, will produce excellent economic and social benefits.

The rich nutrients in hawthorn seeds and leaves should not be overlooked. China is rich in hawthorn resources and the prospects for developing new medicines or producing hawthorn leaf products would be very promising if this renewable resource could be fully utilised. It is important to look at hawthorn chemistry as a whole. In the future, it is important to continue to explore the nutritional and health benefits of hawthorn and to develop value-added foods and supplements based on the functional components of hawthorn, based on extracts and active substances from other parts of hawthorn.

## Figures and Tables

**Figure 1 foods-11-02861-f001:**
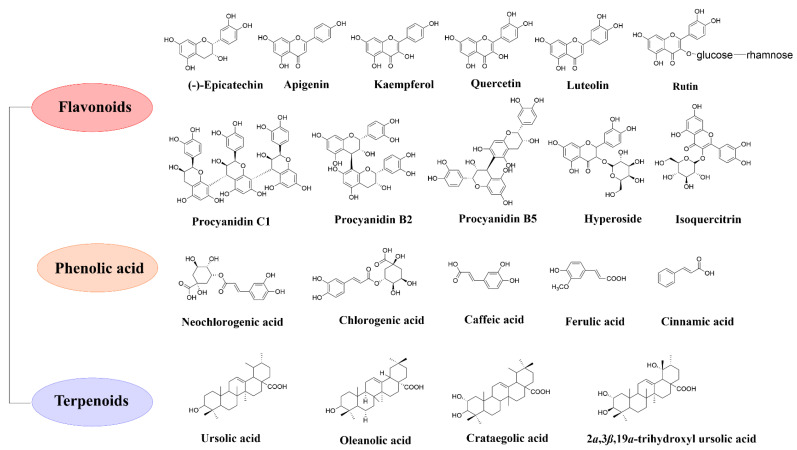
The main bioactive compounds of hawthorn.

**Figure 2 foods-11-02861-f002:**
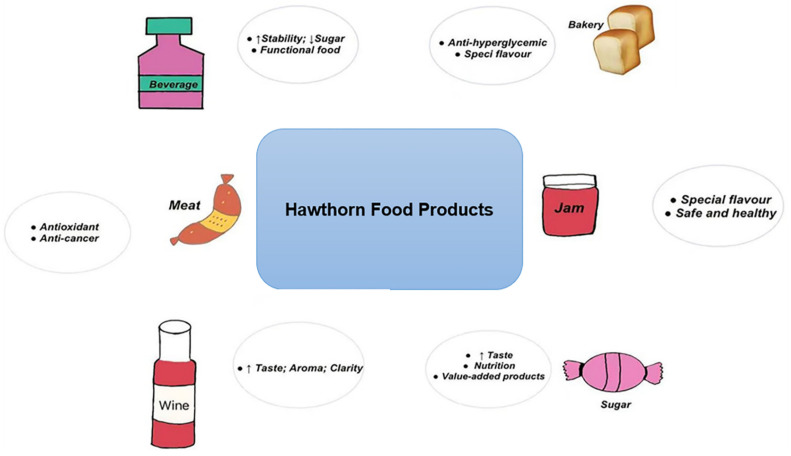
Modern food applications of hawthorn.

**Figure 3 foods-11-02861-f003:**
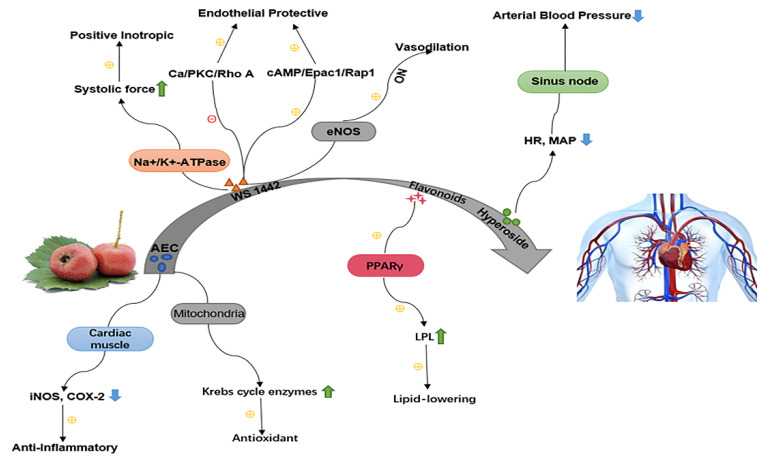
Mechanism of action of hawthorn in the protection against cardiovascular disease. (1). Alcoholic extract of *Crataegus oxyacantha* (AEC) pretreatment maintained mitochondrial antioxidant status and prevented mitochondrial lipid per-oxidative damage and decrease in Krebs cycle enzymes induced by isoproterenol in rat heart.; AEC can act on myocardial tissue to reduce iNOS expression and downregulate COX-2 to exert anti-inflammatory effects. (2). Hawthorn extract WS1442 increases contractility and increases positive muscle strength by inhibiting the Na^+^/K^+^-ATPase pump; WS1442 acts at the serine 1177 site to phosphorylate eNOS and increase NO-mediated vasodilatation; WS1442 effectively protects the vascular endothelium by inhibiting the Ca/PKC/Rho A pathway and activating the cAMP/Epaci/Rap1 pathway. (3). Hawthorn flavonoids exert hypolipidemic effects by acting on the PPAR*γ* pathway to increase LDL expression in blood vessels. (4). High doses of hawthorn ginsenoside can cause significant reductions in heart rate (HR) and mean arterial pressure (MAP), and induce sinus node.

**Table 1 foods-11-02861-t001:** Nutritional composition of hawthorn in different regions. Values are expressed as a percentage (%) and as g per 100 g fresh weight.

Species		*C. Monogyna* Jacq. Var. *Monogyna*		*C. pinnatifida* Bunge		NM		*C. pinnatifida*	
Country	Source	Turkey	Wild	China	Wild	China	Wild	China	Wild
Protein		3.03%		0.7		0.5		3.14%	
Water		68.98%		ND		73		77.48%	
Fat		ND		0.2		0.6		1.3%	
Pectin		ND		3–4		ND		13	
Energy(kJ)		ND		ND		397		364	
Dietary fiber		ND		ND		3.1		33%	
Ca		0.1		ND		52		0.06	
K		1.6		ND		299		1.02	
Fe		6.2		2.1		0.9		0.003	
Na		5.7		68		5.4		0.005	

ND: not determined.

## Data Availability

Not applicable.

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
