# Peer review of "Food Applications and Potential Health Benefits of Hawthorn"

_foods, 2022, doi:10.3390/foods11182861_

Round 1

Reviewer 1 Report

This review manuscript and its content is interesting and useful for the researcher and industries dealing with foods. But there are several issues in this manuscript which should be addressed to make the manuscript more informative .

1.      In the abstract (line 14) mention the name of a few important bioactive compounds from Hawthorn, having nutraceutical properties.

2.      Flavonoids, terpenoids, and phenols (line 17) are the broad classes of bioactive substances that are harbored by most plants. I strongly recommend adding the name of a few key members of each family that are abundantly found in Hawthorn and have some nutritional and medicinal value.

3.      Keywords must be selected more wisely. Do not use keywords such as hawthorn; food application; or health benefits that are already present in the title.

4.      The introduction needs to rewrite thoroughly emphasizing the need for the review of Hawthorn Food Applications and Health Benefits. The author also describes how the present review is extending the knowledge of the articles published on similar topics.

       i.      https://doi.org/10.4028/www.scientific.net/AMM.140.350

  ii.      https://doi.org/10.1016/j.jff.2022.104988

5.      It would be better if the author can merge the plant characteristic with the introduction section.

6. Write the appropriate units such as mg/g or g/kg etc., for the numerical values, indicated in “Section 3“(lines 74, 75, and 76) and “section 4” (lines 137,138). Furthermore, make a comparison table for the nutritive values of Hawthorn from different regions. 

7.      Write the full name of the abbreviation while appearing first in the article e.g Gal A and DE (line 90). 

8.      In table 1 for the compound name “Hesperidin” extraction solvent 50% (1:10 m:v) what does the mean of 1:10 m:v for extraction solvent, address the issue?

9.      It would be better and easy to understand if the author can provide a combined table for the health benefits of different compounds/extracts of Hawthorn, their effective concentration, and their molecular target.

10.      Is the statement correct “Hawthorn isolated compounds……a promising drug for the treatment of melanoma” (Line No. 309-311).

11.  What are compound 12, compound 1~4 section 7.4?

12.  I will suggest that author can make a combined figure representing the molecular event exerted by Hawthorn isolated compounds responsible for various health benefits. 

13.  Finally, the author should provide some research direction as the future perspective which is lacking in the conclusion part.

14. whole manuscript need to be thoroughly check for grammatical and punctuation errors. 

Reviewer 2 Report

The manuscript (“Food Applications and Potential Health Benefits of Hawthorn”), touches on a very current issue of functional foods from hawthorn. The problem undertaken at work is interesting, however, in the manuscript there are some places that must be revised. Title clearly describes what the manuscript is about. Abstract adequately describes the work. Cited references not always correct. There is no information about the figures and tables included in the work. Moreover, the summary should be changed. Consequently, I think minor revision study is necessary.

Reviewer's suggestions below:

L. 27-28: “People have shifted from drugs to wild medicinal plants, traditional Chinese herbs and "ready-to-use therapeutic foods" [1,2].” - in my opinion, not all people chose this way rather some of them  

Regarding “Plant Characteristics”

The content of the chapter does not completely correspond to the title. The chapter should be included only the plant characteristics.

Regarding "4. Chemical Composition of Hawthorn" - My suggestion is to combine chapter 4 with chapter no 3. Nutritional Value" and change the title.

Regarding Figure 1. - information on literature sources is missing. Please provide the reference.

Regarding Table 1. - The amount of individual flavonoids in hawthorn is important. Please complete the table.Similarly in Table 2

Figure 2 should be deleted.

Regarding Figure 3.: The wine picture is incorrect (this is rather a beer). Should be changed.-

Moreover, the author of the drawing (source of the content) and/or the literature sources should be indicated below the picture

L.174-180: - information on literature sources is missing. Please provide the reference.

L.247-248:” „Li et al. used 1.15% hawthorn leaf flavonoids……flavonoid flavor [59].” should be replaced with: „Li et al. [59]used 1.15% hawthorn leaf flavonoids……flavonoid flavor.”

L267:” Okanagan-Colville; and the use of..” should be replaced with: “Okanagan-Colville and the use of..”

L.654- 657 – information on literature sources is missing. Please provide the reference.

Regarding conclusion:

The summary should concern only the facts presented in the article (e.g. whether the authors in the paper described heads in fermented dairy products or hawthorn paste, etc.) and future directions of development. Please correct the summary.

Work needs to be adapted to the editorial requirements of the “Foods” journal. For example, the Authors write: [72,73] and in another place they write: [17], [52].

Round 2

Reviewer 1 Report

The review has been revised and I recommend for the publication of the article.